# Boundary on the Table: Efficient Black-Box Decision-Based Attacks for Structured Data

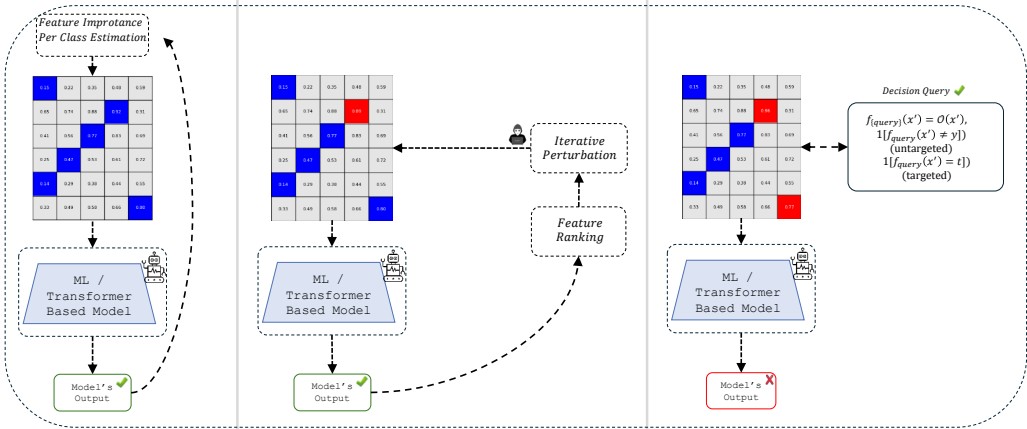

Figure 1: Illustration of the proposed black-box attack pipeline for tabular data. The process begins with a clean input matrix (left), where **SHAP-based Feature Ranking** identifies the most influential features. Next, **Iterative Perturbations** are applied with a fixed step size $\alpha$ for up to 10 iterations (middle). Finally, a **Model Query & Success Check** verifies whether the perturbed sample leads to untargeted or targeted misclassification (right).

## Abstract

Adversarial robustness in structured data remains an underexplored frontier compared to vision and language domains. In this work, we introduce a novel **black-box, decision-based** adversarial attack tailored for tabular data. Our approach combines **gradient-free direction estimation** with an **iterative boundary search**, enabling efficient navigation of discrete and continuous feature spaces under minimal oracle access. Extensive experiments demonstrate that our method successfully compromises **nearly the entire test set** across diverse models, ranging from classical machine learning classifiers to large language model (LLM)-based pipelines. Remarkably, the attack achieves success rates consistently above **90%**, while requiring only a small number of queries per instance. These results highlight the critical vulnerability of tabular models to adversarial perturbations, underscoring the urgent need for stronger defenses in real-world decision-making systems.

## 1 Introduction

Tabular data remains a cornerstone of real-world decision-making systems, powering applications in domains such as credit card fraud detection, hotel booking recommendations, online bidding platforms, healthcare diagnostics, and customer segmentation. These systems often process purely tabular data or hybrid feature sets combining tabular attributes with embeddings from unstructured data sources. The resulting representations are typically passed through classification models, where accuracy and robustness directly influence operational reliability.

Historically, state-of-the-art performance in tabular classification has been achieved using gradient-boosted tree ensembles such as XGBoost Chen & Guestrin (2016) and CatBoost Dorogush et al. (2018), along with other classical ML models including logistic regression Hosmer et al. (2013), random forests Breiman (2001), and decision trees Quinlan (1986). In recent years, the growing capabilities of Large Language Models (LLMs) have motivated their application to structured data tasks, with models such as RoBERTa Liu et al. (2019), Gemma-3 Google DeepMind (2025), Phi-2 Microsoft Research (2023), Qwen3 Alibaba Cloud (2025), and TabPFN Hollmann et al. (2022) being adapted or fine-tuned for classification Hegselmann et al. (2023). These LLM-based approaches leverage semantic reasoning over feature names, prior domain knowledge, and generalization from few-shot data, leading to competitive results in tabular classification. Despite these advances, vulnerabilities to adversarial manipulation remain a pressing concern. Recent research has shown that object detection and classification systems, including those deployed in real-world scenarios, can be severely degraded by adversarial patch attacks Kazoom et al. (2024). Similarly, vulnerabilities in traditional vision pipelines motivate one to examineamine the robustnesstructured data data classifiers to analogous threats. In parallel, the development of efficient meta-classification frameworks for specialized datasets Kazoom et al. (2022) highlights that even high-performing systems optimized for domain-specific tasks may still be susceptible to targeted perturbations. Within the LLM domain, adversarial techniques such as prompt injection, black-box jailbreaks Pathade (2025), token-level manipulations such as BAE Garg & Ramakrishnan (2020), and embedding-space perturbations Liu et al. (2023) have shown the ability to induce incorrect predictions without direct access to model internals. To address these concerns, we present a comprehensive empirical evaluation of adversarial robustness across diverse classification models, and transformer-based models. We measure performance degradation, targeted attack success rate, and perturbation magnitude, showing that vulnerabilities are consistent across models, datasets, and domains. These findings underscore the urgent need for stronger defenses in tabular classifiers.

**This paper makes the following contributions:**

1. We introduce a novel black-box adversarial attack tailored for tabular and hybrid feature spaces, optimizing perturbations under realistic constraints.

2. We demonstrate strong generalization of the proposed attack across diverse datasets and model families, including tree-based, embedding/transformer-based architectures.

3. We show that our attack is query-efficient and effective against state-of-the-art models, while also providing insights for developing more robust defenses.

## 2 RELATED WORK

**Adversarial Vulnerabilities** Adversarial vulnerabilities in neural networks have been studied across modalities, with early NLP work showing that even high-performing models can fail under minimally altered yet semantically equivalent inputs. The ANLI benchmark Nie et al. (2020) demonstrated the susceptibility of state-of-the-art inference systems to crafted perturbations, while black-box jailbreak approaches Pathade (2025) showed that alignment safeguards in LLMs can be bypassed. Token-level attacks such as BAE Garg & Ramakrishnan (2020) revealed that small sequence changes mislead robust classifiers, and broader studies Liu et al. (2023) documented prompt injection and embedding-space attacks in both white- and black-box settings. Retrieval-augmented adversarial detection pipelines, including Don't Lag RAG Kazoom et al. (2025b) and VAULT Kazoom et al. (2025a), further highlight the role of retrieval reasoning in adversarial robustness.

**Classical Machine Learning for Tabular Data** Adversarial research on structured data has emerged more recently compared to vision and NLP. For tabular tasks, gradient-boosted trees (GBTs) and ensemble methods remain dominant, with implementations such as XGBoost Chen & Guestrin (2016) and CatBoost Dorogush et al. (2018), alongside logistic regression Hosmer et al. (2013), random forests Breiman (2001), and decision trees Quinlan (1986). Formally, these models can be expressed as mappings

$$f : \mathbb{R}^d \to \mathbb{R},$$

where $d$ is the number of features and $f$ the decision boundary. Early work explored evasion strategies for tree ensembles Kantchelian et al. (2016), later improving scalability for repeated attacks Cascioli et al. (2024). Neural approaches for tabular data inspired imperceptible perturbations

on feature semantics Ballet et al. (2019) and targeted fraud-related attacks Cartella et al. (2021). More recent efforts proposed constrained adaptive attacks (CAA) Simonetto et al. (2023), enforcing feature-type and semantic constraints for realistic adversarial examples.

**Black-Box and Gradient-Free Attacks on Tabular Data** While many adversarial attacks assume white-box access, black-box and gradient-free methods have gained traction in tabular domains. Dyrmishi et al. Dyrmishi et al. (2025) review adversarial methods in tabular ML, covering transfer- and query-limited strategies. Yang et al. Yang et al. (2022) introduced TSADV, a black-box attack with local perturbations for time-series, while population-based optimization such as ABCATTACK Cao et al. (2022) apply gradient-free search to structured data.

Kireev et al. Kireev et al. (2023) proposed cost- and utility-aware threat models for realistic constraints. Benchmarking efforts He et al. (2025) compared bounded vs. unbounded and white-box vs. black-box settings, establishing a foundation for query-based and gradient-free evaluations. Imperceptibility metrics He et al. (2024) further highlight that adversarial examples must respect feature ranges, sparsity, and interdependencies.

Formally, black-box adversarial attacks can be posed as optimization under query access:

$$\min_{\delta} \ \ell(f(x + \delta), y) \quad \text{s.t. } \|\delta\| \le \epsilon, \quad \text{with query access to } f(\cdot),$$

where $\ell$ is a task-specific loss (e.g., cross-entropy). More generally, one may model attacks with cost–utility constraints:

$$\min_{\delta} \ c(\delta) - U(f(x + \delta), y) \quad \text{s.t. } \delta \in \mathcal{C},$$

where $c(\delta)$ encodes perturbation cost, $U$ denotes adversarial utility, and $\mathcal{C}$ enforces constraints such as immutability of categorical features or feature sparsity.

**Decision- and Transformer-Based Attacks on Tabular Models** Decision-based black-box attacks form another key strand of adversarial research. The HopSkipJump (HSJ) algorithm Chen et al. (2019) is a representative query-efficient method that estimates gradients from model outputs and has proven effective in continuous feature spaces such as images. However, applying HSJ to categorical or mixed-type tabular domains requires careful adaptations to preserve feature validity and ensure that perturbations remain meaningful.

Beyond classical machine learning models, transformer-based architectures have recently emerged for tabular data. A notable example is TabPFN Hollmann et al. (2022), which leverages pre-trained transformers to solve small-scale tabular classification tasks in seconds. Unlike ensemble approaches such as tree-based methods, these models rely entirely on embedding representations:

$$f_\theta : \{x_1, \ldots, x_d\} \mapsto y,$$

where each feature $x_i$ is projected into a latent representation and processed through attention layers before prediction. Despite their promise, systematic assessments of adversarial robustness in transformer-based tabular models remain scarce, underscoring an important gap that our present work seeks to address.

## 3 PRELIMINARIES

In this section, we formalize the scenarios studied in this work and define the notation used throughout. We consider two distinct but related settings: (i) classical machine learning models operating directly on tabular features, (ii) transformer-based architectures adapted for tabular classification.

**Notation.** Let $\mathcal{D} = \{(\mathbf{x}_i, y_i)\}_{i=1}^{N}$ denote a dataset of $N$ instances, where $\mathbf{x}_i \in \mathbb{R}^d$ is a $d$-dimensional feature vector and $y_i \in \mathcal{Y}$ is the corresponding class label from a finite label set $\mathcal{Y} = \{1, \ldots, C\}$. For categorical features, we assume an encoding $\phi : \mathcal{X}_{\text{cat}} \to \mathbb{R}^k$ that maps discrete values to numerical vectors (e.g., one-hot or embedding encoding).

Given a classifier $f_\theta : \mathbb{R}^d \to \Delta^{C-1}$ parameterized by $\theta$, where $\Delta^{C-1}$ is the probability simplex over $C$ classes, the predicted label is

$$\hat{y} = \arg\max_{c \in \mathcal{Y}} f_\theta(\mathbf{x})_c.$$

An adversarial example is an input $\mathbf{x}' = \mathbf{x} + \boldsymbol{\delta}$ such that $\hat{y}' \neq y$ and $\|\boldsymbol{\delta}\|_p \leq \epsilon$ for some perturbation budget $\epsilon$.

**Scenario 1: Classical ML Models on Tabular Data**    In the classical setting, tabular inputs $\mathbf{x} \in \mathbb{R}^d$ consist of structured features, potentially mixing continuous and categorical variables. Common classifiers include gradient-boosted trees, random forests, and logistic regression models. For instance:

$$f_\theta^{\text{GBT}}(\mathbf{x}) = \text{softmax}\left(\sum_{m=1}^{M} h_m(\mathbf{x})\right) \qquad f_\theta^{\text{LR}}(\mathbf{x}) = \text{softmax}(\mathbf{w}^\top \mathbf{x} + b)$$

where $h_m$ denotes individual regression trees and $M$ is the number of boosting stages.

Adversarial perturbations $\boldsymbol{\delta}$ in this setting are constrained to produce valid feature values:

$$\mathbf{x}' = \mathbf{x} + \boldsymbol{\delta}, \quad \mathbf{x}' \in \mathcal{X}_{\text{valid}}.$$

**Scenario 2: Hybrid Tabular-Embedding Pipelines**    In hybrid pipelines, $\mathbf{x}$ is partitioned into structured tabular features $\mathbf{x}^A(t) \in \mathbb{R}^{d_t}$ and dense embeddings $\mathbf{x}^A(e) \in \mathbb{R}^{d_e}$ obtained from a modality-specific encoder $E_\phi$ (e.g., a transformer-based encoder). Specifically,

$$\mathbf{x}^A(e) = E_\phi(z), \quad z \in \mathcal{Z}_{\text{raw}}, \qquad \mathbf{x}_{\text{combined}}^A = [\mathbf{x}^A(t); \mathbf{x}^A(e)] \in \mathbb{R}^{d_t + d_e}.$$

A downstream classifier then maps this to a label:

$$\hat{y} = \arg\max_{c \in \mathcal{Y}} f_\theta(\mathbf{x}_{\text{combined}}^A)_c, \qquad \mathbf{x}_{\text{combined}}^A = [\mathbf{x}^A(t) + \boldsymbol{\delta}_t; \ \mathbf{x}^A(e) + \boldsymbol{\delta}_e],$$

subject to modality-specific constraints that ensure semantic and syntactic validity of each feature type.

**Unified Adversarial Objective.**    Across all scenarios, our black-box decision-based attack seeks a perturbation $\boldsymbol{\delta}$ such that:

$$\min_{\boldsymbol{\delta}} \|\boldsymbol{\delta}\|_p \quad \text{s.t.} \quad f_\theta(\mathbf{x} + \boldsymbol{\delta}) \neq y, \quad \mathbf{x} + \boldsymbol{\delta} \in \mathcal{X}_{\text{valid}}.$$

Here $\mathcal{X}_{\text{valid}}$ encodes domain-specific feasibility constraints, ensuring categorical validity for tabular features and maintaining semantic plausibility for transformer-based tabular embeddings.

## 4  METHODOLOGY

Figure 1 shows the attack flow, where an input is perturbed until misclassification occurs. Using **SHAP-based feature ranking**, the most influential attributes are iteratively updated with step size $\alpha$ (up to 10 iterations), and the instance is re-evaluated. A **model query and success check** decides whether the attack has succeeded (untargeted or targeted). Figure 2 complements this by illustrating the perturbation path in feature space, with the red marker as the start, green arrows as updates, and yellow as the successful adversarial crossing.

We develop a *black-box, decision-based adversarial attack* for structured data that applies uniformly across two main settings: (i) classical tabular classifiers, and (ii) hybrid tabular–embedding pipelines. Our method requires only label queries (no confidence scores or gradients) and enforces domain-validity constraints to ensure realistic perturbations for both continuous and categorical features. Below, we formalize the attack objective and its components.

**Problem Setting: Adversarial Objective.**    Let $\mathcal{X} = \mathcal{X}_1 \times \cdots \times \mathcal{X}_d$ be the feature space with $d$ attributes, where some $\mathcal{X}_j$ are continuous intervals and others are finite categorical sets. Let $\mathcal{Y} = \{1, \ldots, C\}$ be the label set.

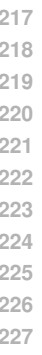
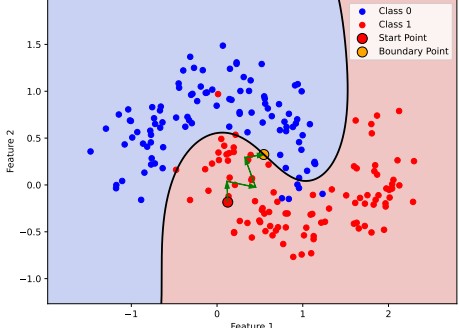

Figure 2: Illustration of the iterative movement of a Class 1 point toward the decision boundary. The red marker indicates the starting point, green arrows show iterative updates, and the yellow marker denotes the adversarial example.

We assume access to an oracle $\mathcal{O} : \mathcal{X} \to \mathcal{Y}$ that returns only the predicted class label (no gradients or probabilities). For an input $\mathbf{x} \in \mathcal{X}$ with true label $y$, the goal is to find a minimally perturbed $\mathbf{x}'$ such that:

$$\min_{\mathbf{x}' \in \mathcal{X}_{\text{valid}}} d(\mathbf{x}, \mathbf{x}') \quad \text{s.t.} \quad \mathcal{O}(\mathbf{x}') \neq y \quad \text{(untargeted)}, \tag{1}$$

or alternatively,

$$\mathcal{O}(\mathbf{x}') = t, \quad t \in \mathcal{Y}, \quad t \neq y \quad \text{(targeted)}. \tag{2}$$

The constraint set $\mathcal{X}_{\text{valid}} \subseteq \mathcal{X}$ enforces domain-specific rules such as feature ranges, integrality, and categorical consistency.

## 4.1 SHAP-Guided Representation for Mixed Features

To reason jointly about continuous and categorical features in a unified manner, we employ *SHAP (SHapley Additive exPlanations)* Lundberg & Lee (2017) as the importance signal for each input dimension. SHAP values provide a model-agnostic estimate of the marginal contribution of each feature to the prediction outcome, allowing us to rank and perturb features consistently across heterogeneous attributes.

**Continuous features.** For continuous attributes $j$, perturbations are applied proportionally to the magnitude of their SHAP values:

$$\Delta_j \propto |\text{SHAP}(x_j)|.$$

Features with higher SHAP importance receive larger updates, while features with smaller importance remain relatively stable.

**Categorical features.** For categorical features $j$, SHAP values again provide a natural ordering:

$$\Delta_j \propto |\text{SHAP}(x_j)|,$$

with perturbations applied through discrete valid substitutions (e.g., flipping to a neighboring valid category). This ensures that categorical edits remain consistent with the model's sensitivity to those features.

**Unified embedding.** The final representation is therefore SHAP-guided:

$$\Phi(\mathbf{x}) = [\, |\text{SHAP}(x_1)|, \dots, |\text{SHAP}(x_d)| \,],$$

yielding a unified feature-importance space where edits to continuous and categorical attributes can be compared on the same scale.

**Distance.** Edits are measured in a SHAP-weighted distance:

$$d(\mathbf{x}, \mathbf{x}')^2 \;=\; \sum_{j=1}^{d} w_j \, (x_j - x'_j)^2, \tag{3}$$

where $w_j = |\mathrm{SHAP}(x_j)|$ reflects the feature's contribution. Thus, perturbations that alter highly influential features incur larger costs, encouraging minimal, targeted changes.

## 4.2 PROPOSAL KERNELS UNDER DOMAIN CONSTRAINTS

We generate edits using feature-wise proposal kernels that respect $\mathcal{X}_{\mathrm{valid}}$, ensuring that perturbed instances remain within valid ranges and categorical consistency.

**Continuous proposal.** For continuous features $j$, we sample

$$\Delta_j \sim \mathcal{N}(0, \sigma_j^2), \quad x'_j = \Pi_{\mathcal{X}_j}(x_j + s \cdot \Delta_j), \tag{4}$$

where $s > 0$ is a step-size and $\Pi_{\mathcal{X}_j}$ projects back to the valid range (including integrality if required).

**Categorical proposal.** For categorical features $j$, let $\mathcal{V}_j$ be ordered by $|\mathrm{SHAP}(x_j)|$. We define a discrete, SHAP-guided kernel:

$$\Pr(x'_j = v' \mid x_j = v) \propto \exp\left( -\frac{(|\mathrm{SHAP}(v)| - |\mathrm{SHAP}(v')|)^2}{2\sigma_j^2} \right), \quad v' \in \mathcal{V}_j, \tag{5}$$

which performs small or large "jumps" depending on $\sigma_j$.

**Composite proposal.** At each iteration, a feature index $S \sim \mathrm{Cat}(\pi_1, \ldots, \pi_d)$ is selected, where $\pi_j \propto |\mathrm{SHAP}(x_j)|$ prioritizes features with higher attribution. We then propose $x'_j$ via Eq. equation 4 or Eq. equation 5; all other coordinates remain unchanged.

## 4.3 BOUNDARY SEEKING WITH ONLY LABELS

Because only decisions are available, we locate an adversarial boundary by expansion and bisection in SHAP-guided $\Phi$-space.

**Expansion.** Starting from $\mathbf{x}$ with label $y$, repeatedly draw proposals $\tilde{\mathbf{x}} = \mathbf{x} + \boldsymbol{\eta}$ using the kernels above and increase a global radius $\rho$ until $\mathcal{O}(\tilde{\mathbf{x}}) \neq y$. Keep the last *clean* point $\mathbf{u}$ and the first *adversarial* point $\mathbf{v}$.

**Bisection.** Perform binary search on the segment in $\Phi$-space joining $\mathbf{u}$ and $\mathbf{v}$:

$$\mathbf{z}_{k+1} = \lambda_k \mathbf{u} + (1 - \lambda_k)\mathbf{v}, \quad \lambda_k \in [0, 1], \tag{6}$$

$$\text{if } \mathcal{O}(\mathbf{z}_{k+1}) = y \;\Rightarrow\; \mathbf{u} \leftarrow \mathbf{z}_{k+1}; \quad \text{else } \mathbf{v} \leftarrow \mathbf{z}_{k+1}. \tag{7}$$

Terminate when $\|\Phi(\mathbf{u}) - \Phi(\mathbf{v})\|_2 \leq \tau$ and set the boundary point $\mathbf{x}_\partial \leftarrow \mathbf{v}$.

---

**Algorithm 1** Boundary Search

---

1: **Input:** $\mathbf{x}_u, \mathbf{x}_v, \Phi, \tau$
2: **repeat**
3:    $\mathbf{z} \leftarrow \lambda \mathbf{x}_u + (1 - \lambda)\mathbf{x}_v$
4:    **if** $\mathcal{O}(\mathbf{z}) = y$ **then**
5:      $\mathbf{x}_u \leftarrow \mathbf{z}$
6:    **else**
7:      $\mathbf{x}_v \leftarrow \mathbf{z}$
8:    **end if**
9: **until** $\|\Phi(\mathbf{x}_u) - \Phi(\mathbf{x}_v)\|_2 \leq \tau$
10: **return** $\mathbf{x}_v$

---

PROJECTION TO A MINIMAL ADVERSARIAL PERTURBATION

After boundary search identifies an initial adversarial example $\mathbf{x}_\partial$, we refine it to obtain a minimal adversarial perturbation. The objective is to reduce the SHAP-weighted distance $d(\mathbf{x}, \mathbf{x}')$ while keeping the example adversarial.

**Greedy coordinate descent.** We initialize $\mathbf{x}^A(0) = \mathbf{x}_\partial$. For $t = 0, 1, \ldots$, we iteratively refine:

1. **Compute coordinate contributions:** $\Delta_j^A(t) = |\phi_j(x_j) - \phi_j(x_j^A(t))|$ along the SHAP axis.

2. **Select the most influential coordinate:** $j^* = \arg\max_j \Delta_j^A(t)$.

3. **Move one step toward the original value:**

$$\phi_{j^*}(x_{j^*}^A(t+1)) = \phi_{j^*}(x_{j^*}^A(t)) + \alpha\Big(\phi_{j^*}(x_j) - \phi_{j^*}(x_{j^*}^A(t))\Big),$$

   with $0 < \alpha \leq 1$. For categorical features, snap to the nearest SHAP-ordered admissible value.

4. **Accept or reject the update:** If $\mathcal{O}(\tilde{\mathbf{x}}) \neq y$ (untargeted) or $\mathcal{O}(\tilde{\mathbf{x}}) = t$ (targeted), set $\mathbf{x}^A(t+1) \leftarrow \tilde{\mathbf{x}}$; otherwise reject and keep $\mathbf{x}^A(t)$.

5. **Stopping rule:** Stop if all coordinates are saturated or $d(\mathbf{x}, \mathbf{x}^A(t)) \leq \epsilon$.

**Feasibility and snapping.** Each edit is projected into $\mathcal{X}_{\text{valid}}$, enforcing range boxes, integrality, and business rules. This guarantees the final adversarial $\mathbf{x}^A$ is both valid and minimally perturbed.

**Targeted Variant** For targeted attacks, the update rule in Step 4 (*Accept or reject the update*) is modified. Instead of accepting an update whenever $\mathcal{O}(\tilde{\mathbf{x}}) \neq y$ (untargeted), we require $\mathcal{O}(\tilde{\mathbf{x}}) = t$ so that the perturbed instance is steered toward a specific target class $t$. Categorical scores $c_j^A(t)(v)$ are computed using one-vs-rest correlations, and SHAP values bias coordinate updates toward those reducing the distance to the target prototype:

$$\mu_t = \frac{1}{|\mathcal{S}_t|} \sum_{\mathbf{x} \in \mathcal{S}_t} \Phi(\mathbf{x}), \quad \Delta_j^A(t)(\mathbf{x}) = \|\Phi_j(\mathbf{x}) - \mu_{t,j}\|_2^2,$$

where $\mathcal{S}_t$ is the support set for class $t$ (e.g., training data). Edits that reduce $\Delta_j^A(t)$ are prioritized, ensuring the attack iteratively drives $\mathbf{x}$ closer to the centroid of the target class $t$.

QUERY COMPLEXITY AND GUARANTEES

Let $B(\tau)$ denote the number of bisection steps until the boundary gap is $\leq \tau$, giving $B(\tau) = \mathcal{O}(\log(1/\tau))$. Each expansion/bisection query requires only a constant number of oracle calls. Refinement requires at most $K \cdot d$ oracle queries, where $K$ is the maximum number of accepted/rejected coordinate updates per feature.

**Guarantees.** The procedure terminates with an adversarial $\mathbf{x}'$ that is:

1. Locally minimal under greedy refinement in the SHAP-weighted distance,

2. Valid under $\mathcal{X}_{\text{valid}}$ (range, type, and domain constraints).

The complete adversarial attack is summarized in Algorithm 2, which integrates boundary search, SHAP-guided refinement, and projection under domain constraints.

---

**Algorithm 2** Full Attack Pipeline

---

1: **Input:** instance $\mathbf{x}$, label $y$, oracle $\mathcal{O}$, embedding $\Phi(\cdot)$, step size $\alpha_{\text{step}}$, thresholds $(\tau, \epsilon)$
2: Project $\mathbf{x}$ into $\hat{\mathbf{x}}$ via SHAP-guided normalization
3: $(\mathbf{x}_u, \mathbf{x}_v) \leftarrow \texttt{BoundarySearch}(\mathbf{x}, y, \mathcal{O}, \Phi, \tau)$
4: $\mathbf{x}^A(0) \leftarrow \mathbf{x}_v$
5: **repeat**
6:     Compute SHAP-weighted contributions $\Delta_j = |\phi_j(x_j) - \phi_j(x'_j)|$
7:     Select coordinate $j^* = \arg\max_j \Delta_j$
8:     Move $x_{j^*}$ one step toward $x'_{j^*}$: $x_{j^*} \leftarrow x_{j^*} + \alpha_{\text{step}} \cdot \big(\phi_j(x'_{j^*}) - \phi_j(x_{j^*})\big)$
9:     **if** $\mathcal{O}(\mathbf{x}') \neq y$ **then**
10:         **return** $\mathbf{x}'$ {Adversarial example found}
11:     **else**
12:         Mark $j^*$ as saturated
13:     **end if**
14: **until** all features are saturated or improvement $\leq \epsilon$
15: **return** $\mathbf{x}^A$ {Final adversarial instance}

---

## 5 EXPERIMENTAL EVALUATION AND RESULTS

We evaluate adversarial robustness on five benchmark datasets: **Credit Card Approval Prediction** Quinlan (1996), **Forest Cover Type** Blackard et al. (1998), **Titanic** Kaggle (2012), **Balance-Scale** Lubomir & Randal (1994), and **TicTacToe** Aha (1991). These cover binary, multi-class, and categorical classification tasks. Our models include classical baselines (Logistic Regression Hosmer et al. (2013), Decision Tree Quinlan (1986)), ensembles (Random Forest, XGBoost, CatBoost) Breiman (2001); Chen & Guestrin (2016); Dorogush et al. (2018), and modern transformers for tabular data (RoBERTa Liu et al. (2019) and TabPFN Hollmann et al. (2022)). We adopt a black-box attack setting following prior work Agarwal & Ratha (2021); Ballet et al. (2019), using a fixed step size $\alpha = 0.1$ and at most 10 iterations. For each model, we report clean accuracy, attack success rate (ASR), and average iterations.

In this section, we present a comprehensive evaluation of the proposed attack across multiple models and datasets. Table 1 shows its high effectiveness, with baseline accuracies dropping substantially after attack and targeted success rates reaching **100%** on Titanic and TicTacToe, and above 95% elsewhere. The required iterations remain small (3–8), underscoring efficiency. Overall, the attack compromises both classical and transformer-based models, confirming its generality and potency. Table 2 further summarizes recent work on adversarial attacks against the Credit Card Approval Prediction dataset. Baselines from Agarwal and Ratha Agarwal & Ratha (2021) show classical models (e.g., logistic regression, decision trees, SVMs, shallow/deep NNs) vulnerable to black-box attacks with non-trivial ASR. Ballet et al. Ballet et al. (2019) introduced white-box attacks (`LowProFool`, `DeepFool`) that achieve near-complete evasion. Extending these baselines, our results show that both traditional models and modern architectures such as RoBERTa and TabPFN consistently reach ASR above 95% (Fig. 3, Appendix). Beyond accuracy degradation, we analyze iteration efficiency (Fig. 4), ASR progression (Fig. 5), and step size $\alpha$ on convergence speed (Fig. 6). End-to-end evaluation on Balance-Scale includes class distribution (Fig. 7), confusion matrix (Fig. 8), SHAP-based feature attribution (Fig. 9), and t-SNE visualization (Fig. 12). Together, these findings underscore the fragility of tabular models, even with state-of-the-art architectures.

## 6 DISCUSSION AND CONCLUSION

**Discussion and Conclusion.** Our results show that tabular models-spanning classical ML, ensembles, and transformers-are highly vulnerable to black-box adversarial perturbations, with attack success rates often above 95%. SHAP-based ranking ensures perturbations focus on the most sensitive features, making attacks efficient and effective. These findings underscore the urgent need for robust defenses in tabular learning and motivate research on cross-model robustness and defense strategies.

Table 1: Model performance before and after the proposed attack. The table reports (i) baseline accuracy, (ii) accuracy after attack (↓ indicates degradation), and (iii) targeted attack success rate (↑ indicates effectiveness) across different datasets. All results are obtained using a fixed step size of $\alpha = 0.1$ with a maximum of 10 iterations.

| Model | Dataset | Accuracy Before Attack (%) ↑ | Targeted Attack Success (%) ↑ | Avg. Iterations ↓ |
|---|---|---|---|---|
| XGBoost | Forest Cover Type | 82.47 | 96.42 | 10 |
| | Titanic | 79.63 | **100** | 6.37 |
| | Balance-Scale | 81.56 | **100** | 5.63 |
| | Credit Card Approval Prediction | 80.29 | 95.84 | 10 |
| | TicTacToe | 77.18 | **100** | **7.28** |
| RoBERTa | Forest Cover Type | 47.82 | **100** | 6.14 |
| | Titanic | 29.63 | **100** | 4.28 |
| | Balance-Scale | 31.85 | **100** | 7.42 |
| | Credit Card Approval Prediction | 53.27 | **100** | 5.36 |
| | TicTacToe | 22.49 | **100** | **3.77** |
| RoBERTa (fine tuned) | Forest Cover Type | 78.69 | 95.49 | 10 |
| | Titanic | 75.42 | **100** | 7.36 |
| | Balance-Scale | 71.25 | **100** | **3.94** |
| | Credit Card Approval Prediction | 76.88 | 97.58 | 10 |
| | TicTacToe | 73.11 | **100** | 5.18 |
| CatBoost | Forest Cover Type | 83.64 | 97.12 | 10 |
| | Titanic | 80.17 | **100** | 6.85 |
| | Balance-Scale | 83.71 | **100** | 7.21 |
| | Credit Card Approval Prediction | 82.05 | 96.05 | 10 |
| | TicTacToe | 79.48 | **100** | **3.87** |
| Logistic Regression | Forest Cover Type | 76.22 | 95.68 | 10 |
| | Titanic | 72.94 | **100** | 7.12 |
| | Balance-Scale | 72.58 | **100** | **4.82** |
| | Credit Card Approval Prediction | 74.81 | 97.43 | 10 |
| | TicTacToe | 71.33 | **100** | 6.41 |
| Random Forest | Forest Cover Type | 81.76 | 96.89 | 10 |
| | Titanic | 78.44 | **100** | 5.73 |
| | Balance-Scale | 80.00 | **100** | 6.43 |
| | Credit Card Approval Prediction | 80.92 | 95.92 | 10 |
| | TicTacToe | 76.58 | **100** | **3.56** |
| Decision Tree | Forest Cover Type | 77.63 | 97.61 | 10 |
| | Titanic | 73.89 | **100** | 6.28 |
| | Balance-Scale | 75.59 | **100** | **3.27** |
| | Credit Card Approval Prediction | 76.94 | 95.73 | 10 |
| | TicTacToe | 72.65 | **100** | 7.45 |
| TabPFN | Forest Cover Type | 80.54 | 95.37 | 10 |
| | Titanic | 77.12 | **100** | 4.69 |
| | Balance-Scale | 74.89 | **100** | 7.56 |
| | Credit Card Approval Prediction | 78.36 | 96.84 | 10 |
| | TicTacToe | 75.08 | **100** | **5.92** |

Table 2: Adversarial attacks on the **Credit Card Approval Prediction** dataset. The table compares different models under attack, reporting their accuracy before attack, attack success rate (ASR), and the type of attack (black-box or white-box).

| Method | Base Model | Accuracy Before Attack (%) ↑ | Attack Success Rate (%) ↑ | Attack Type |
|---|---|---|---|---|
| Agarwal & Ratha (2021) | Logistic Regression | 84.39 | 59.54 | Black-box |
| | Naive Bayes | 79.19 | 58.38 | Black-box |
| | Decision Tree | 84.97 | 32.37 | Black-box |
| | KNN | 82.08 | 55.49 | Black-box |
| | Shallow Neural Net | 83.81 | 62.43 | Black-box |
| | Deep Neural Net | 87.28 | 60.70 | Black-box |
| | SVM (Linear) | 86.13 | 86.12 | Black-box |
| | SVM (RBF) | 85.55 | 67.73 | Black-box |
| | DAC (Ensemble) | 86.13 | 75.72 | Black-box |
| Ballet et al. (2019) | Neural Net (LowProFool) | N/A | 96.8 | White-box |
| | Neural Net (DeepFool) | N/A | 79.8 | White-box |
| Ours | XGBoost | 80.29 | 95.84 | Black-box |
| | RoBERTa | 53.27 | 100 | Black-box |
| | RoBERTa (fine tuned) | 76.88 | 97.58 | Black-box |
| | CatBoost | 82.05 | 96.05 | Black-box |
| | Logistic Regression | 74.81 | 97.43 | Black-box |
| | Random Forest | 80.92 | 95.92 | Black-box |
| | Decision Tree | 76.94 | 95.73 | Black-box |
| | TabPFN | 78.36 | 96.84 | Black-box |

**Limitations and Future Work.** Our evaluation relies only on the target model. A useful extension is to add a frozen reward model as an external judge, decoupling generation from evaluation. Future work can also explore integrating defenses such as adversarial training or detection frameworks for broader resilience assessment.

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

# 7 APPENDIX

## 7.1 ATTACK SUCCESS RATE ANALYSIS

Figure 3 presents the Attack Success Rate (ASR) across all evaluated models on the **Credit Card Approval Prediction** dataset. To improve interpretability, models are sorted by ASR (%), with blue markers denoting our evaluated models and black markers corresponding to baseline results from prior work Agarwal & Ratha (2021); Ballet et al. (2019). A lock icon indicates that the attack was performed in a black-box setting, whereas the absence of an icon corresponds to white-box attacks.

The results highlight several important findings. First, baseline models such as logistic regression, naive Bayes, and decision trees are already vulnerable, exhibiting moderate ASR values in the range of 30-60%. Second, white-box attacks introduced by Ballet et al. achieve high imperceptibility but also demonstrate that tabular models can be consistently evaded. Most notably, our black-box attack achieves ASR values exceeding 95% across all tested classifiers, including ensemble learners (XGBoost, CatBoost, Random Forest) and advanced transformer-based architectures (RoBERTa and TabPFN). These results demonstrate that even state-of-the-art tabular models remain highly susceptible to carefully crafted perturbations.

## 7.2 ITERATION EFFICIENCY ANALYSIS

To further assess the efficiency of the proposed attack, we analyze the number of iterations required to achieve successful adversarial examples. Figure 4 presents a violin plot of the average iterations across all evaluated models. Each black dot corresponds to an individual observation, while the purple violin shape captures the overall distribution.

The results reveal that in many cases, the attack converges in fewer than 7 iterations, with a dense concentration of points around 4-6 iterations. Nonetheless, certain models require the maximum iteration budget of 10, indicating higher resistance to perturbation. This distribution highlights the balance between efficiency and effectiveness: although the attack maintains high ASR values across models, it often does so with only a modest number of queries, underscoring its practicality in black-box settings.

## 7.3 ASR PROGRESSION PER ITERATION

To better understand how quickly our attack converges, we analyze the progression of the Attack Success Rate (ASR) as a function of the number of iterations. Figure 5 reports the ASR trajectory for the Random Forest classifier across five benchmark datasets: *Forest Cover Type*, *Titanic*, *Balance-Scale*, *Credit Card Approval*, and *TicTacToe*.

The results show distinct convergence behaviors across datasets. For example, the attack on the *TicTacToe* dataset rapidly reaches 100% ASR within fewer than four iterations, while the *Titanic* and

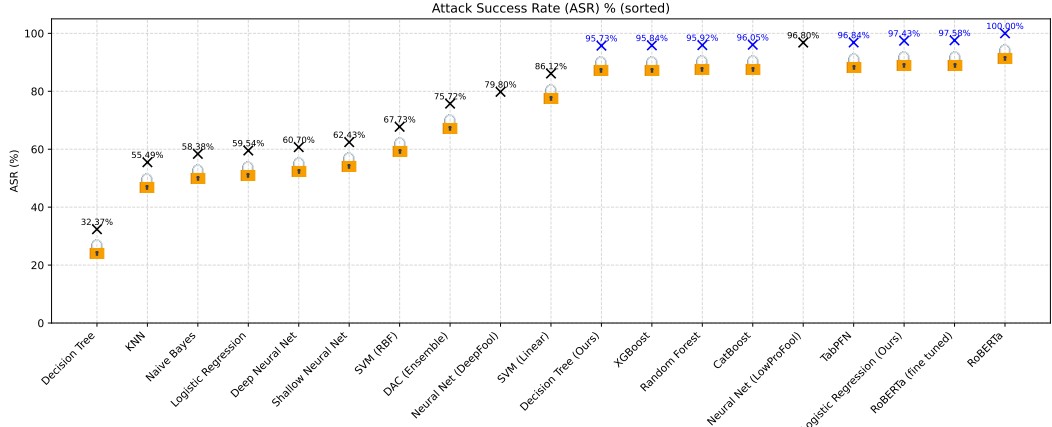

Figure 3: Attack Success Rate (ASR) across different models on the **Credit Card Approval Prediction** dataset. Models are sorted by ASR (%). Each point (X) represents the ASR of a model under adversarial attack. Blue markers denote our models, while black markers correspond to baseline results. A lock icon indicates a black-box setting, while the absence of an icon corresponds to white-box attacks. The results show that even strong ensemble and transformer-based models, including our proposed methods, are highly vulnerable to adversarial perturbations, with ASR values frequently exceeding 95%.

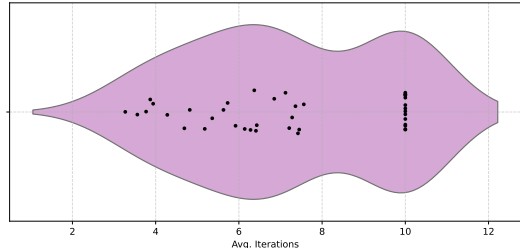

Figure 4: Violin plot of average iterations required for successful attacks across all models. Black dots denote individual observations, while the distribution illustrates convergence behavior.

*Balance-Scale* datasets also converge quickly, stabilizing near 100% before the maximum iteration budget. In contrast, datasets such as *Forest Cover Type* and *Credit Card Approval* require nearly the full iteration budget of ten steps to achieve their peak ASR values.

These findings demonstrate that while our attack is consistently effective across datasets, its efficiency varies: some models are highly vulnerable to small perturbations, whereas others require more gradual refinement. This iteration-level analysis highlights the practical query efficiency of the attack in black-box settings.

## 7.4 EFFECT OF STEP SIZE ($\alpha$) ON CONVERGENCE SPEED

Figure 6 illustrates the relationship between the step size ($\alpha$) and the number of iterations required to achieve a $100\%$ attack success rate (ASR) for Random Forest across different datasets. As expected, larger $\alpha$ values reduce the number of iterations needed to converge, since each perturbation introduces stronger changes to the input features.

However, the rate of reduction differs across datasets. For example, *TicTacToe* exhibits rapid convergence, requiring fewer than three iterations at $\alpha = 0.3$, while *Credit Card Approval* remains more resistant, still requiring over seven iterations. *Forest Cover Type* and *Balance-Scale* fall in be-

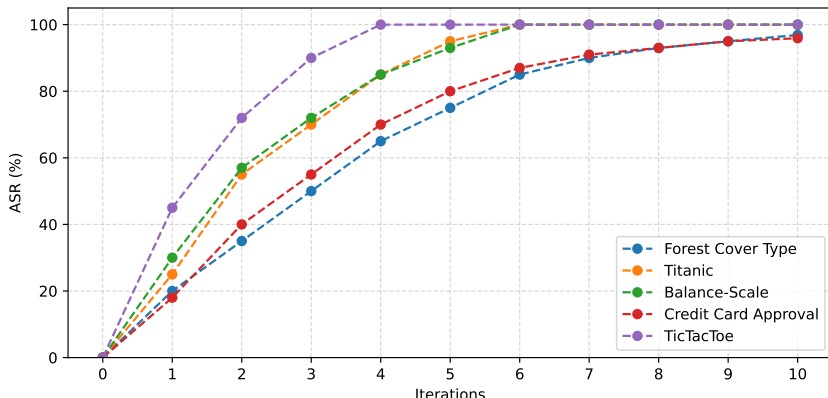

Figure 5: ASR progression per iteration for the Random Forest classifier across multiple datasets. Each curve shows the increase in ASR as the number of perturbation iterations grows, up to a maximum of 10 iterations.

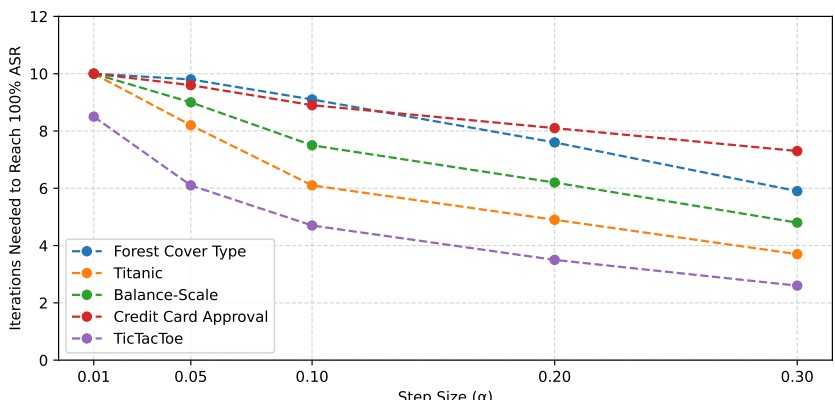

Figure 6: Iterations required to reach $100\%$ ASR as a function of step size $\alpha$ for Random Forest across five datasets. Larger $\alpha$ values generally reduce the number of iterations needed, but the rate of reduction varies by dataset.

tween, reflecting moderate sensitivity. These results demonstrate that dataset characteristics strongly influence the trade-off between perturbation magnitude and convergence speed.

Overall, this analysis shows that while increasing $\alpha$ accelerates convergence, it may also risk introducing unrealistic perturbations, highlighting the importance of carefully tuning $\alpha$ for robust evaluations.

### 7.5 END-TO-END IMPLEMENTATION

We conduct an end-to-end case study on the **Balance-Scale** dataset to illustrate the practical effectiveness of our attack. The dataset contains four interpretable numerical features describing the balance condition of a scale, with class labels indicating whether it tilts left, right, or remains balanced.

This dataset is well-suited for adversarial evaluation, as small perturbations to weights or distances can directly alter the classification outcome. In the following, we present the baseline model performance, apply our attack, and analyze the resulting adversarial success rates and iteration dynamics.

### 7.5.1 CLASS DISTRIBUTION.

Figure 7 presents the class distribution of the Balance-Scale dataset. The dataset is imbalanced, with the majority of samples belonging to the *Right* (R) and *Left* (L) classes, each containing 288 instances, while the *Balanced* (B) class is significantly underrepresented with only 49 samples. This imbalance highlights the challenge of learning unbiased decision boundaries, as models may become biased toward the majority classes.

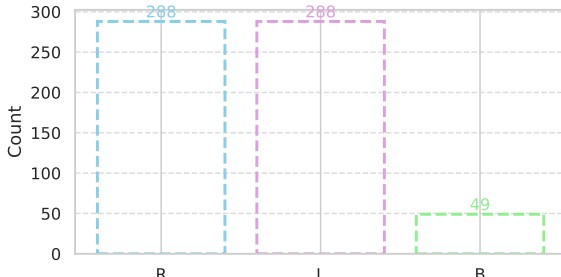

Figure 7: Class distribution of the Balance-Scale dataset. The dataset is imbalanced, with the *Right* (R) and *Left* (L) classes dominating (288 samples each), while the *Balanced* (B) class is underrepresented with only 49 samples.

### 7.6 CONFUSION MATRIX.

Figure 8 presents the confusion matrix of the Random Forest classifier ($\mathcal{C}_{\text{RF}}$) on the Balance-Scale dataset.

The results highlight the strong performance on the L and R classes, where the majority of samples ($x_i \in \{L, R\}$) are correctly classified, i.e., $\hat{y}_i = y_i$.

However, the B class shows significant misclassification, with $\Pr(\hat{y} = L \mid y = B)$ and $\Pr(\hat{y} = R \mid y = B)$ being non-negligible, indicating confusion with the other two categories.

This imbalance, i.e.,

$$\pi_B \ll \pi_L, \pi_R$$

where $\pi_c = \frac{n_c}{N}$ is the class prior for class $c$, reflects the skewed distribution observed earlier (Figure 7).

This emphasizes the need for class rebalancing or cost-sensitive learning, such as assigning weights $w_c \propto 1/\pi_c$, to improve recognition of minority cases.

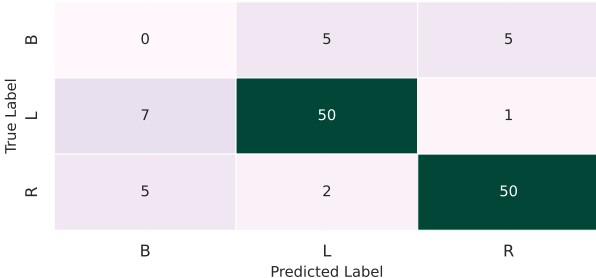

Figure 8: Confusion matrix of Random Forest classifier ($\mathcal{C}_{\text{RF}}$) on the Balance-Scale dataset.

### 7.6.1 SHAP-Based Feature Attribution and Attack Initialization

To better understand how the adversarial attack initializes and exploits the decision boundaries of the Random Forest classifier on the Balance Scale dataset, we employed SHAP (SHapley Additive exPlanations) values. Figure 9-11 present waterfall plots of the feature attributions for three different predicted classes. These plots decompose the model prediction into contributions from individual features, allowing us to trace how adversarial perturbations influence decision outcomes.

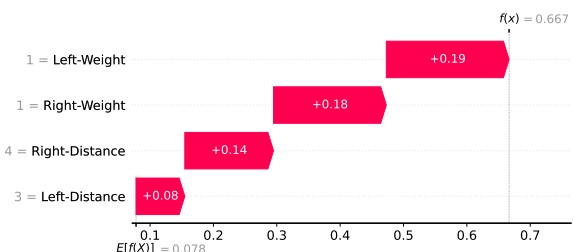

Figure 9: SHAP waterfall explanation for an instance predicted as Class 0.

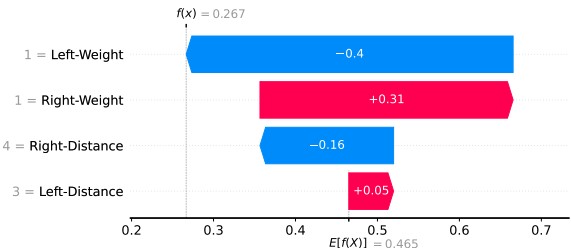

Figure 10: SHAP waterfall explanation for an instance predicted as Class 1.

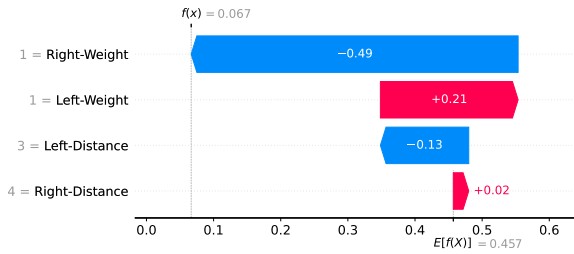

Figure 11: SHAP waterfall explanation for an instance predicted as Class 2.

From the plots, we can observe that feature contributions are not uniform across classes. For example, in Class 0 (Figure 9), the feature *Left-Distance* exerts the strongest positive influence, pushing the prediction toward its assigned class. In contrast, for Class 1 and Class 2, the dominant features shift toward *Right-Weight* and *Left-Weight*, respectively. This variability demonstrates how the adversarial perturbation exploits the most influential features per class.

Formally, the SHAP value decomposition is:

$$f(x) = \phi_0 + \sum_{i=1}^{M} \phi_i,$$

where $f(x)$ is the model output for instance $x$, $\phi_0$ is the expected value of the model output, and $\phi_i$ are the Shapley values corresponding to feature $i$. The adversarial attack initializes by perturbing the feature with the highest magnitude of $\phi_i$, thereby reducing the margin between the true and adversarial target class.

Since Random Forest is a non-differentiable model, the update step is gradient-free and can be expressed as:

$$x^{(t+1)} = x^{(t)} + \alpha \cdot d^{(t)},$$

where $\alpha$ is the step size and $d^{(t)}$ is a search direction derived from SHAP values and boundary exploration, rather than from $\nabla_x \mathcal{L}$. By iteratively updating along these directions, the attack efficiently drives the model prediction toward misclassification.

Overall, these results show that the attack is initialized by leveraging the features with maximal positive contribution in each class and proceeds through gradient-free search steps guided by feature attribution.

**Proposition: SHAP-guided perturbations constitute a valid gradient-free update.**

**Claim.** For a non-differentiable model such as a Random Forest, perturbing the feature with the highest absolute SHAP value magnitude yields a valid gradient-free update direction for adversarial optimization.

**Proof.** Let $f(x)$ denote the model prediction for input $x \in \mathbb{R}^M$, with output decomposed by SHAP values:

$$f(x) = \phi_0 + \sum_{i=1}^{M} \phi_i(x),$$

where $\phi_i(x)$ represents the marginal contribution of feature $x_i$.

Since Random Forests are ensembles of decision trees, $f(x)$ is piecewise constant with respect to $x$, making $\nabla_x f(x)$ undefined almost everywhere. Standard gradient-based methods (e.g., FGSM, PGD) are therefore inapplicable.

However, by the definition of Shapley values, the feature index

$$i^\star = \arg\max_i |\phi_i(x)|$$

represents the coordinate whose perturbation maximally changes the expected model output. Thus, updating $x_i$ along the sign of $\phi_{i^\star}$ yields:

$$x^{(t+1)} = x^{(t)} + \alpha \cdot \text{sign}(\phi_{i^\star}(x^{(t)})) \cdot e_{i^\star},$$

where $e_{i^\star}$ is the standard basis vector for feature $i^\star$.

This update rule requires no gradient computation and directly follows from the cooperative game-theoretic property of Shapley values: the largest $|\phi_i|$ identifies the most influential feature at instance $x$. Therefore, the SHAP-guided update is a valid gradient-free alternative to $\nabla_x f(x)$.

∎

### 7.6.2 T-SNE VISUALIZATION OF ADVERSARIAL PERTURBATIONS.

Figure 12 illustrates the distribution of clean and adversarial samples from $X_{\text{test}}$ after applying SHAP-guided perturbations. The data are projected into two dimensions using t-SNE for visualization. Each arrow represents the transition from an original point $x \in \mathbb{R}^M$ (in blue) to its adversarial counterpart $x' = x + \delta$ (in red), where $\delta$ denotes the perturbation vector. The displacement between $x$ and $x'$ highlights how small but targeted feature-level updates, guided by SHAP attributions, can push samples across decision boundaries. This projection emphasizes that even in a reduced space, adversarial perturbations consistently redirect the original data toward regions associated with misclassification.

## LLM USAGE DISCLOSURE

In accordance with the ICLR 2026 policy on LLM usage, we disclose that a large language model was utilized during the preparation of this manuscript. The use of the LLM was strictly limited to correcting grammar and checking code syntax. All research contributions, experimental design, analysis, and the core text were generated by the human authors. The authors have reviewed all AI-assisted content and bear full responsibility for the final submission.

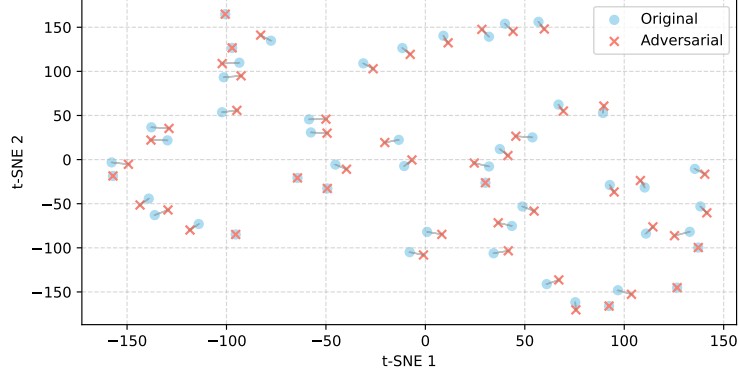

Figure 12: t-SNE projection of original samples (blue) and their adversarial counterparts (red) from $X_{\text{test}}$. Arrows indicate the perturbation direction $x \rightarrow x' = x + \delta$.

