# OpenReview forum: "Boundary on the Table: Efficient Black-Box Decision-Based Attacks for Structured Data"
_ICLR.cc/2026/Conference — ICLR 2026 Conference Withdrawn Submission_

### Official Review · Reviewer_FT8J · 2025-10-29

**Soundness:** 2
**Presentation:** 2
**Contribution:** 3
**Rating:** 2
**Confidence:** 5

**Summary:**

This paper introduces a novel black-box, decision-based adversarial attack specifically tailored for structured (tabular) data. The core contribution is a two-stage pipeline: first, it employs SHAP (SHapley Additive exPlanations) to create a model-agnostic, unified feature importance ranking that handles both continuous and categorical features. Second, guided by this ranking, it uses an iterative boundary search algorithm (combining expansion and bisection) to efficiently find a minimal perturbation that causes the target model to misclassify an input. The authors conduct experiments across five datasets and a wide range of models—from classical classifiers to modern transformers—demonstrating high attack success rates (often >95%) with a small number of search iterations.

**Strengths:**

1. The paper addresses a critical and relatively under-explored area: the adversarial robustness of machine learning models on tabular data. As these models are widely deployed in high-stakes domains like finance and healthcare, understanding their vulnerabilities is of paramount importance. This work makes a timely and significant contribution by focusing on this problem.

2. The primary methodological novelty lies in using SHAP values as the guiding signal for a black-box attack. This provides an elegant and intuitive way to unify the treatment of heterogeneous feature types (continuous and categorical) into a single importance-guided search space. This approach circumvents the need for gradient estimation and is well-suited for the decision-based setting.

3. The authors have performed a comprehensive experimental evaluation. The breadth of models tested—spanning logistic regression, tree ensembles (XGBoost, CatBoost), and transformer-based architectures (RoBERTa, TabPFN)—is commendable and provides strong evidence for the generalizability of the proposed attack. The consistent high performance across multiple diverse datasets further strengthens these claims.

4. The paper is well-written, clearly structured, and easy to follow. The proposed attack pipeline is illustrated effectively in Figure 1, and the overall methodology is explained logically. The abstract and introduction do an excellent job of motivating the problem and outlining the paper's contributions.

**Weaknesses:**

1. The paper fails to clearly articulate why a new, specialized attack for tabular data is necessary in the decision-based black-box setting. Tabular data is a mix of continuous and discrete features. However, the paper does not explain the unique challenges this mixture presents that could not be solved by simply combining existing attack strategies for continuous domains and discrete domains. Without this crucial justification, the contribution risks being perceived as an incremental combination of existing ideas rather than a novel solution to a distinct problem.

2. The central claim of high query efficiency is based on a flawed accounting of query costs. The analysis exclusively focuses on the low number of iterations in the boundary search phase while completely ignoring the query cost required to compute the SHAP values. In a true black-box setting, obtaining model-agnostic SHAP values (e.g., via KernelSHAP) is computationally expensive and requires a substantial number of model queries, often scaling with the number of features. By omitting this significant upfront cost, the paper presents a misleading picture of the attack's true efficiency. This omission makes it impossible to fairly compare the method against other attacks.

3. The experimental evaluation is critically lacking a direct, head-to-head comparison with established black-box attack baselines. While Table 2 references results from prior work, this is insufficient as experimental settings can vary significantly. To validate the superiority of the proposed method, a direct comparison against a strong, relevant baseline like HSJA, adapted for tabular data, is essential. Without such a comparison on total query cost and perturbation magnitude under an identical experimental framework, the paper's claims of state-of-the-art performance are unsubstantiated.

Minor but Notable Presentation Issues:
- The font size in several key figures, particularly Figure 1 and Figure 2, is too small to be comfortably read, hindering the paper's clarity.
- There are instances of confusing and inconsistent notation, like x^A_combined used in line 181 and line 186.

**Questions:**

1. Can you please provide a full accounting of the query complexity of your attack? This should include the queries needed for the initial SHAP value estimation plus the queries for the boundary search. How does this total query cost compare empirically to a baseline like HSJA on the same task?

2. What are the specific, fundamental challenges of attacking mixed-type tabular data in a decision-based setting that are not adequately addressed by a straightforward combination of existing continuous-space and discrete-space search strategies?

3.  Your method relies on a generic X_valid constraint set. How would your SHAP-guided approach handle more complex real-world constraints, such as (a) immutable features (e.g., gender, race) that should never be perturbed, and (b) correlated features where changing one necessitates a valid change in another (e.g., Age and Years of Experience)?

4. While you demonstrate a high attack success rate, the paper lacks a quantitative analysis of the resulting perturbation magnitude. Could you provide statistics on the norms of the adversarial perturbations generated by your attack? This would allow for a more complete comparison of its effectiveness and stealthiness.

---

### Official Review · Reviewer_m7C5 · 2025-10-31

**Soundness:** 2
**Presentation:** 2
**Contribution:** 3
**Rating:** 4
**Confidence:** 4

**Summary:**

The paper introduces a decision-based black-box adversarial attack that can be applied to a wide range of tabular models including ensemble gradient boosting, neural networks, and tabular foundation models. The method leverages explainability outputs from the SHAP framework to obtain feature rankings and magnitudes, which then guide iterative updates toward targeted or untargeted misclassifications through three phases: bisection, expansion, and greedy search.
The algorithm is designed to minimize perturbation magnitude while maintaining continuous ranges and categorical consistency constraints. It is recognized for its efficiency and fast convergence in terms of iterations. The authors evaluate the method across diverse datasets from domains such as finance, gaming, and geography.

**Strengths:**

- Originality: The paper proposes a novel way to utilize SHAP values through an iterative process involving expansion, bisection, and greedy coordinate descent.
- Clarity: The paper is generally clear and easy to follow.
- Quality: The authors conduct evaluations using multiple datasets and model families. The study addresses realistic constraints inherent to tabular data.
- Impact: The work has potential significance for both security and interpretability in machine learning on tabular data. The proposed attack strategy could have important implications for robustness assessment and defense development.

**Weaknesses:**

Some relevant prior works that also use SHAP-based feature importance for adversarial evasion on tabular data appear to have been overlooked, and clarifying how this paper’s approach differs from or advances beyond them would strengthen the discussion of novelty: Khazanchi, V., Kulkarni, P., Govindarajulu, Y., & Parmar, M. (2024). MISLEAD: Manipulating Importance of Selected features for Learning Epsilon in Evasion Attack Deception. arXiv preprint arXiv:2404.15656.

- The average number of oracle calls used throughout the entire process of the attack is missing in the experimental setup section. This inclues all queries required starting from SHAP value estimation arriving to expansion, bisection and greedy coordinate descent.

- Performance should  be compared (e.g., in terms of success rate and oracle calls) with stronger baselines such as: Simonetto, T., Ghamizi, S., & Cordy, M. (2024). Constrained Adaptive Attack: Effective Adversarial Attack Against Deep Neural Networks for Tabular Data. Advances in Neural Information Processing Systems, 37, 27817-27849.

- Some methodology details could be better specified, for instance, how constraints are handled in Section 4. Rather than a general statement (“we always ensure constraints are respected”), explicit mathematical expressions improves the readability and interpretation.

- Results presentation could be improved. The comparative tables and figures make it difficult to directly compare same models evaluated across strategies. For example, Table 2 and Figure 3 could be reorganized for clearer cross-model comparisons.

**Questions:**

1.	Could you clarify what you mean by “integrality” and “business rules constraints” in Section 4? Please specify how these are formulated or enforced.
2.	In Step 3 (“Project to minimal adversarial perturbation → Greedy coordinate descent”), is the update applied in the input space or in the SHAP value space? If it is in SHAP space, how are those changes projected back into the input space?
3.	What is the average number of oracle calls required to compute the SHAP value representations across datasets (before starting the adversarial process)?

---

### Official Review · Reviewer_bFuf · 2025-11-01

**Soundness:** 2
**Presentation:** 3
**Contribution:** 2
**Rating:** 2
**Confidence:** 3

**Summary:**

This paper proposes a new black-box, decision-based adversarial attack designed specifically for tabular data. The proposed method works with minimal oracle access. The attack operates by first using SHAP-based feature ranking to identify the most influential features. It then employs an iterative boundary search to find the decision boundary, followed by a SHAP-guided greedy coordinate descent algorithm to find a minimal perturbation.
The authors test their attack against a wide range of models. The experiments show that the attack achieves extremely high success rates (ASR > 95%) with a small number of queries across all tested models and datasets.

**Strengths:**

The paper conducts a comprehensive experiment across diverse models while gaining promising ASRs, suggesting that tabular models are far more vulnerable to decision-based attacks than perhaps previously thought.

**Weaknesses:**

1. The paper’s central claim of being decision-based (implying label-only access) appears to be in direct conflict with its core methodology, which relies heavily on SHAP values. However, this paper lacks explaination on how these SHAP values are obtained in a decision-only setting.

2. The core algorithmic components are not particularly novel. The boundary search (Section 4.3) is essentially a standard binary search, and the refinement (Page 7) is a greedy coordinate descent. The main novelty is the use of SHAP values to guide this process.

3. The paper fails to compare HopSkipJump (Chen et al., 2019, as mentioned in line 133 of the paper), which is a required baseline for this paper that claims a new decision-based attack. The comparisons in Table 2 are insufficient, as they don't appear to include modern decision-based attackers.

**Questions:**

Apart from addressing the concerns in the Weaknesses section, an additional question lies below.

The ASRs of >95-100% are exceptionally high, far exceeding other black-box methods in Table 2. Could you comment on whether this is due to your method’s superiority or a difference in experimental setup (e.g., perturbation budget, query limits, or the flawed assumption of having SHAP values)?

---

### Note · Authors · 2025-11-13

I have read and agree with the venue's withdrawal policy on behalf of myself and my co-authors.